# Surface protein glycosylation conserved in the human pathogen *Mycoplasma genitalium* and retained in the synthetic organism JCVI-Syn3A

**John William Sanford**[1], **James Mobley**[2], **Kevin Dybvig**[1], **Thomas Prescott Atkinson**[1], **James Michael Daubenspeck**[1]*

**1** Department of Pediatrics, University of Alabama at Birmingham, Birmingham, Alabama, United States of America, **2** Department of Anesthesiology and Perioperative Medicine, University of Alabama at Birmingham, Birmingham, Alabama, United States of America

* jametex@uab.edu

## Abstract

Protein glycosylation has been reported in all forms of life. The genus *Mycoplasma* is composed of highly genome-streamlined bacterial symbionts, making them model organisms for investigating minimal genome concepts. Previous work from our group showed mycoplasmas scavenge hexoses from exogenous oligosaccharides to glycosylate surface proteins at serine, threonine, asparagine, and glutamine residues without utilizing a consensus sequence as seen in canonical glycosylation systems. We report here that this surface protein hexosylation system is conserved in *Mycoplasma genitalium*, a human urogenital pathogen with a 580-kbp genome that can be cultured axenically. We also report this modification is found in the ruminant pathogen *Mycoplasma mycoides* subsp. *capri* and is conserved in JCVI-Syn3A, a nonpathogenic mycoplasma with a synthetic minimal *M. mycoides* genome containing genes that are essential for survival and robust growth under axenic culture conditions. In contrast to known glycoproteins, we have detected evidence of glycosylation of aspartic acid and glutamic acid residues, which expands the pool of potential glycosyl acceptors in bacteria to include the acidic amino acids.

## Introduction

Mollicute metabolism primarily revolves around parasitizing host cells for nutrients and evading the host immune response. Mollicutes are bacterial symbionts that lack a cell wall and produce persistent infection/colonization in their hosts by employing sophisticated immune evasion strategies, including dynamic surfaceomes with phase-varied proteins and proteolytic processing [1]. Our group has previously reported a surface protein glycosylation system in the murine pathobionts *Mycoplasma arthritidis* and *M. pulmonis*. This glycosylation system is distinct from other bacterial glycosylation systems. Hexoses are attached through both *N*- and

**Data availability statement:** The mass spectrometry proteomics data have been deposited to the ProteomeXchange Consortium [45] via the PRIDE [46] partner repository with the dataset identifier PXD057584 and 10.6019/PXD057584.

**Funding:** The author(s) received no specific funding for this work.

**Competing interests:** authors have declared that no competing interests exist.

*O*-linkages to asparagine, glutamine, serine, and threonine residues. Isotope labeling experiments indicate the hexose is cleaved from exogenous oligosaccharides as precursors to the glycosylation reaction [2], instead of the nucleotide sugars utilized by Leloir glycosyltransferases [3]. We have previously proposed that the energy from the glycosidic bond is utilized by an unidentified enzyme that catalyzes this hexosylation reaction. This hexosylation system has a high degree of variability with no discernable amino acid consensus sequence flanking the modification site. This results in a wide population of glycoproteins on the surface of *M. arthritidis* and *M. pulmonis* cells [4].

Bacteria in the class *Mollicutes* are often used as model organisms for exploring the questions regarding the concept of minimal life. Most mycoplasmas have genomes under one megabase pair in length due to genome streamlining as an adaptation to colonizing and parasitizing their eukaryotic hosts. *Mycoplasma genitalium* (Mgen) is a human urogenital pathogen that has streamlined its 580-kbp genome under natural selective pressures resulting in the smallest genome of any identified organism able to be maintained in axenic culture. Despite Mgen's small genome encoding a dearth of *de novo* metabolic pathways, the organism has a polar organelle for gliding motility and attachment to host cells [5], produces extracellular polymeric substances to form biofilms [6], and expresses a universal immunoglobulin-binding protein [7]. We report here that Mgen possesses the hexosylation system previously reported in the murine pathobionts.

Developed in 2016 by the J. Craig Venter Institute, JCVI-Syn3A (Syn3A)'s 543-kbp genome was synthesized based on a near-minimal gene set from its *M. mycoides* parent organism, a virulent pathogen of ruminants. This bottom-up synthetic genome reduction has resulted in the organism having its virulence completely attenuated, even being unable to parasitize host cells *in vitro* [8–10]. Syn3A's 493 genes include 19 that are quasi-essential for robust growth and consistent cell cycle and 149 annotated to have unknown functions [11]. The presence of a gene or metabolic pathway retained in Syn3A's artificial genome strongly implies an essential physiological function under axenic culture conditions for mollicutes.

After identifying hexosylation in Mgen, we analyzed Syn3A and its natural parent organism, *M. mycoides* subsp. *capri* (Mmc) for evidence of hexosylation. Using SDS-PAGE staining for proteins and glycoproteins, we excised bands and confirmed the presence of hexoses in the glycoproteins utilizing gas chromatography-mass spectrometry (GC-MS). Finally, we performed Orbitrap tandem mass spectrometry and analyzed the datasets for hexosylated proteins. Our results indicate that the surface protein hexosylation system is conserved in Mgen, Mmc, and Syn3A with a surprising degree of amino acid diversity. In addition to the *N*-linked modification of asparagine and glutamine, these species also have modified *O*-linked serine, threonine, and tyrosine residues. Unexpectedly, our data show evidence of these species glycosylating aspartic acid and glutamic acid, which has not been described previously in other bacteria.

## Methods

### Strains and preparation

Mgen G37 [12], Mmc GM12 [13], and JCVI-Syn3A [14] were grown in SP4 growth medium to stationary phase as previously described [15], with 3 mg/mL of maltose added to the medium to act as the required oligosaccharide substrate for mycoplasma hexosylation. After washing three times with PBS, whole cell lysates were used for SDS-PAGE analysis and GC-MS. Our previous experiments in *M. pulmonis* and *M. arthritidis* showed no evidence of glyco-staining in cytoplasmic material [16], leading us to utilize membrane-enriched samples for glycoproteomic analysis in the current study. To enrich for membrane material for Orbitrap MS, whole cells from all three species were suspended in 2X PBS with Halt Protease inhibitor (Thermofisher Catalog #78430) to prevent any protein degradation. After overnight incubation at 4°C, the samples were lysed by injection with 100 volumes of ultrapure $H_2O$. The lysates were centrifuged at 5,000 x *g* for 10 minutes to remove insoluble material and any unlysed cells before the resulting supernatant was ultracentrifuged at 36,000 x *g* for one hour at 4°C to pellet membrane material [16,17].

### SDS-PAGE

Protein concentration was determined with a BCA assay in triplicate in a 96-well plate format (Thermofisher Catalog #A55865). 25 µg of protein was denatured at 100°C for 10 minutes and separated using precast 10–20% Novex™ Wedgewell Tris-Glycine gels (Thermofisher Catalog #XP10200BOX) run in parallel and stained with Coomassie blue or Pro-Q™ Emerald 300 glycostain (Thermofisher Catalog #P21857).

### GC-MS

Excised SDS-PAGE bands were trypsin digested and treated with acidic methanol to produce the methyl-glycosides. These were derivatized and analyzed by GC-MS as previously described [2].

### Mass spectrometry sample preparation

Rapigest SF (SKU #186001861) was purchased from Waters. Ammonium bicarbonate (AmBic, Catalog #A6141), dithiothreitol (Catalog #D9779), and iodoacetamide (Catalog #I1149) were purchased from Sigma-Aldrich. Trypsin Gold, MS grade (Catalog #PRV5280) was purchased from Promega. Acetonitrile (Catalog #A955), trifluoroacetic acid (Catalog #PI28903), and formic acid (Catalog #A117) were purchased from ThermoFisher. Other materials are listed within the Procedure section below. Rapigest SF was reconstituted in 1 mL of 50 mM AmBic to 0.1% final before use. The membrane pellet was resuspended in the 0.1% Rapigest SF solution and quantified using a BCA protein assay kit (Thermofisher Catalog #PI23225). Twenty micrograms of protein were then digested with trypsin/Lys-C in-solution with reduction (dithiothreitol at 5 mM final) and alkylation (iodoacetamide at 15 mM final), following the manufacturer's protocol. Tryptic digests were cleaned up using Pierce™ C18 Spin Columns (Catalog #89870), and the eluate was finally reconstituted in 5% acetonitrile, 0.1% formic acid at ~125 ng/uL prior to LC-MS analysis.

### Mass spectrometry

Peptide digests (8 µL each) were injected onto a 1260 Infinity nHPLC stack (Agilent Technologies) and separated using a 75-micron I.D. x 15 cm pulled tip C-18 column (Jupiter C-18 300 Å, 5-micron, Phenomenex). This system runs in-line with a Thermo Velos Pro Orbitrap mass spectrometer, equipped with a Nanospray Flex™ ion source (Thermofisher), and all data were collected in CID mode. The nHPLC is configured with binary mobile phases that include solvent A (0.1% formic acid in ddH$_2$O), and solvent B (0.1% formic acid in 15% ddH$_2$O/ 85% acetonitrile), programmed as follows; 10 min at 5% solvent B (2 µL/ min, load), 90 min at 5%−40% solvent B (linear: 0.5 nL/min, analyze), 5 min at 70% solvent B (2 µL/ min, wash), 10 min at 0% solvent B (2 µL/ min, equilibrate). Following each parent ion scan (300–1200 m/z at 60k resolution),

MS$^2$ fragmentation data were collected on the top-most intense 18 ions. The Syn3A and Mmc samples were then repeated using a one- and two-hour gradient and including +1 ions to expand the available peptides for comparison, while Mgen was rerun with a one-hour gradient due to scarcity of material owing to the organism's fastidious nature. For data dependent scans, charge state screening and dynamic exclusion were enabled with a repeat count of 2, repeat duration of 30 s, and exclusion duration of 90 s. The XCalibur RAW files were collected in profile mode.

### Search parameters

Mass spectrometry data were analyzed with PEAKS Studio 11 (Bioinformatics Solutions Inc., Waterloo, ON) and SEQUEST through Scaffold 5 (Proteome Software, Inc., Portland, OR) to identify peptides with potential hexosylation. We adapted a previously described proteome analysis workflow for *Mycoplasma pneumoniae* [18]. Mass spectra were searched against species-specific proteomes for each organism, while common protein contaminants including serum proteins that were not removed after repeated washes were excluded through a universal contaminant protein library [19]. The MS$^1$ mass tolerance was set to 20 ppm while the MS$^2$ mass tolerance was set to 0.5 Da. The spectra were searched for peptides between six to 20 amino acids long. Carbamidomethylation of Cys (15.03 Da) was set as a constant post-translation modification (PTM) while N-terminus acetylation (42.01 Da), oxidation of Met (16.00 Da), and hexosylation of Ser, Thr, Tyr, Gln, and Asn residues (162.05 Da.) were set as variable PTMs, to account for protein sequence variations, PEAKS' SPIDER algorithm was used after PEAKS DB mass spectra assignments. An additional search was performed expanding the pool of hexosylated residues to include Glu and Asp and compared to the first search. Finally, to account for unspecified PTMs, we performed a third search using the PEAKS PTM Finder. Two variable PTMs and three missed protease cleavage events were allowed. We used a 2% false discovery rate when searching these data. PEAKS tests the statistical validity of putative mass spectra assignments through generating decoy peptide sequences against each protein in the database [20]. All data reported here had −10logP ≥ 25.

Scaffold was used to generate and retain only high confidence IDs while also generating normalized spectral counts across all samples for the purpose of relative quantification. The filter cut-off values were set with minimum peptide length of >5 residues, with no MH + 1 charge states, with peptide probabilities of >80% confidence interval, and with the number of peptides per protein ≥2. The protein probabilities were set to a > 99.0% confidence interval, and a false discovery rate <1.0%. Scaffold incorporates the two most common methods for statistical validation of large proteome datasets, the false discovery rate and protein probability [21–23]. Relative quantification across experiments was then performed via spectral counting [24,25], and when relevant, spectral count abundances were then normalized between samples [26].

### Protein structure prediction

Protein sequences from Mgen, Mmc, and Syn3A used in the mass spectrometry datasets were predicted using Alphafold 3 protein structural modeling software [27]. The Alphafold models were analyzed using Mol* Viewer to visualize the tertiary structure of the glycosylation sites [28].

## Results

### SDS-PAGE and GC-MS of mycoplasma proteins

Whole-cell lysates from three mycoplasmas, Mgen, Mmc, and Syn3A, were analyzed by SDS-PAGE (Fig 1). Parallel gels were stained with either Coomassie or Pro-Q™ Emerald 300 Glycostain to determine if glycosylation was active in each species. Proteins from all three species glycostained at intensities as previously observed in other mycoplasma species [2,4]. Two sets of bands from gel regions that have been previously shown to contain modified proteins in *M. arthritidis* and *M. pulmonis* were excised from the Coomassie-stained gel from each organism, as indicated with red arrows (Fig 1), trypsin digested, and analyzed by GC-MS. The Syn3A spectrum of the generated methyl-glycosides from band B are shown (Fig 2). This spectrum

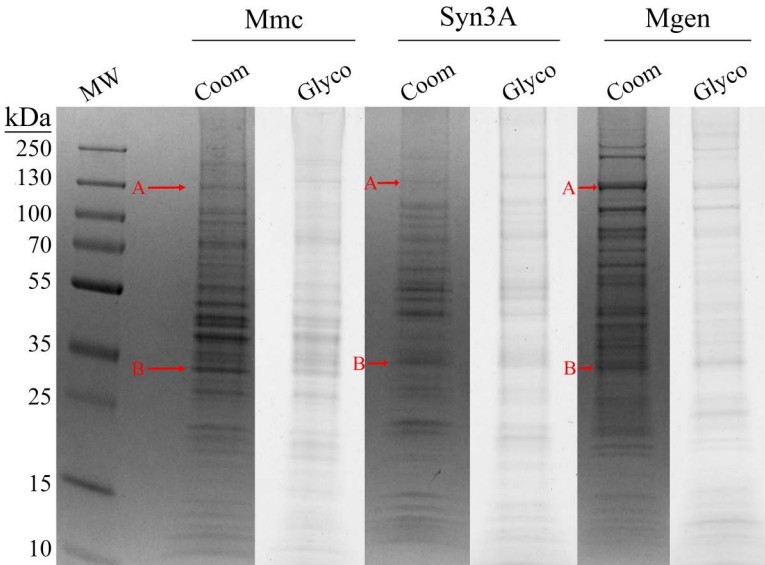

**Fig 1. Polyacrylamide gels of Mmc, Syn3A, and Mgen lysates were stained for protein (Coomassie blue, Coom) and glycan (Pro-Q™ Emerald 300, Glyco) content.** Evidence of hexosylation in all three species was indicated by bands in the glycostain. Arrows indicate bands excised for GC-MS (see Fig 2).

shows a predominant glucose signal along with a small amount of mannose. All analyzed spectra from the A and B bands of each of the three species showed the same two monosaccharides in similar proportions (S2 Fig).

## Hexosylation of mycoplasma proteins

To identify putative glycosylation sites, we focused our efforts on membrane-enriched fractions, as previous work has indicated mycoplasma hexosylation occurs on proteins localized to the membrane but not localized in the cytoplasm [4]. After performing high-resolution LC-MS/MS on in-solution Trypsin/Lys-C digested membrane materials, we searched the mass spectra against the Mgen, Syn3A, and Mmc proteomes in PEAKS 11 software. Our search criteria used a more stringent −10logP cutoff of 25 rather than the value of 20 recommended by PEAKS. PEAKS' peptide search algorithm fuses decoy and target proteins together to reduce the risk of false-positive identification [20]. The mass error tolerance was 20 ppm for the MS$^1$ and 0.5 Da for the MS$^2$ spectra. We used PEAKS 11's PTM Finder and SPIDER algorithms to search for unspecified PTMs and sequence variations. We also searched these data for the addition glycans consisting of permutations of hexoses, amino-hexoses, and deoxyhexoses up to tetrasaccharides. The only instances of glycosylation that were detected were single hexoses. We observed similar ratios of fragmented ion intensities between hexosylated and unmodified peptides for further confidence against false-positive identification. We have identified the five canonical amino acids available as substrates for glycosylation: Ser, Thr, Asn, and Gln, along with the uncommon *O*-linked glycosyl acceptor Tyr. Surprisingly, we also identified hexosylation at the acidic amino acids Asp and Glu (Table 1).

### *N*- and *O*-linked hexosylation of Mgen proteins at canonical amino acids

We detected instances of *O*- and *N*-linked glycosylation in Mgen (Table 1). We analyzed *N*-linked glycosylation sites, including Asn10 of acetate kinase (S3 Fig) on the peptide ILVVN$_{10}$AGSSSIK and Asn$_{45}$ of chaperonin GroEL (S4 Fig) on the peptide AN$_{45}$PLITNDGVTIAK. For acetate kinase, the predicted and observed MS$^1$ mass for the unmodified peptide were both 1188.706. The predicted and observed hexosylated MS$^1$ masses were 1351.767 and 1351.746, respectively with a mass error of −15.535 ppm. The MS$^2$ clearly shows Asn10 of acetate kinase being hexosylated in both the *b*- and

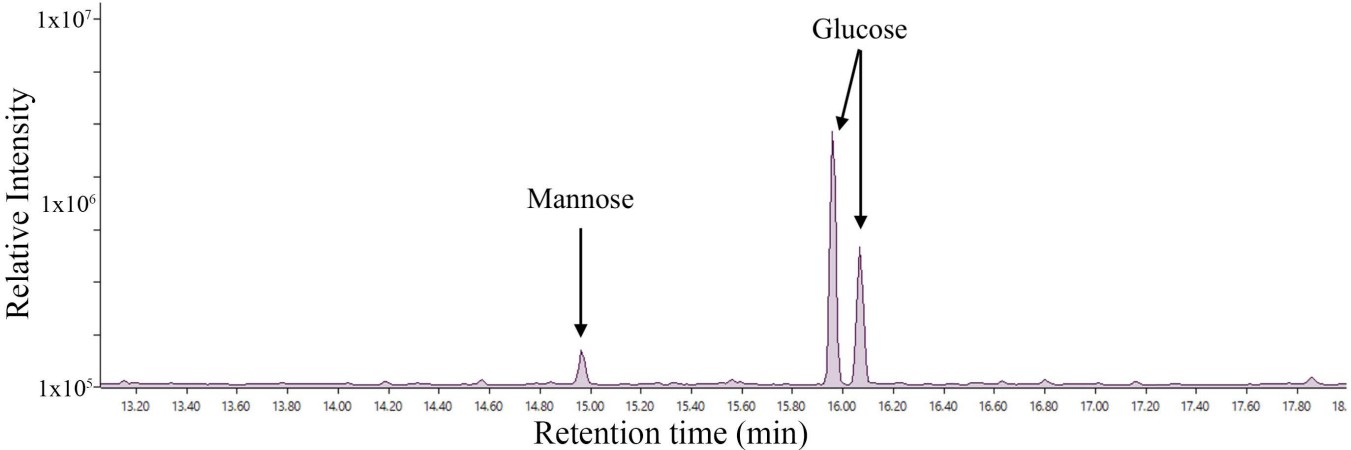

**Fig 2. GC-MS spectrum from a trypsin-digested SDS-PAGE band B from Figure 1 of Syn3A shows glucose is the predominant attached hexose, along with a minor degree of mannose.** This pattern was conserved in all six analyzed bands.

**Table 1. Detected hexosylated Mgen proteins.**

| Accession | Protein Name | Peptide | Observed m/z | Theoretical m/z | z | −10logP |
|---|---|---|---|---|---|---|
| AAC71620.2 | Chaperonin GroEL | AN$_{45}$PLITNDGVTIAK | 815.943 | 815.930 | 2 | 81.93 |
| AAC71566.1 | DNA-directed RNA polymerase subunit beta | HPN$_{319}$DSSLTALELEMENK | 1045.483 | 1045.483 | 2 | 29.56 |
| AAC72471.1 | Elongation Factor Tu | DIAS$_{142}$DEEVQELVAEEVR | 1046.976 | 1046.992 | 2 | 97.85 |
| AAC71307.1 | Elongation Factor G | EFSIKAS$_{302}$DDANFIGLAFK | 712.362 | 712.356 | 3 | 36.30 |
| AAC71523.1 | Glyceraldehyde-3-phosphate dehydrogenase | VS$_{107}$EEGASLHLK | 687.357 | 687.343 | 2 | 100.90 |
| AAC71495.1 | Pyruvate dehydrogenase E1 component subunit beta | EHHSET$_{134}$LEAIYAQIAGLK | 724.711 | 724.699 | 3 | 47.39 |
| AAC71495.1 | Pyruvate dehydrogenase E1 component subunit beta | RGE$_{307}$KYQFEINAR | 558.287 | 558.281 | 3 | 33.57 |
| AAC71582.1 | Acetate kinase | ILVVN$_{10}$AGSSSIK | 450.582 | 450.599 | 3 | 24.12 |
| AAC71521.1 | Phosphate acetyltransferase | VRDPS$_{102}$SLAATLVALK | 582.337 | 582.328 | 3 | 86.33 |
| AAC71521.1 | Phosphate acetyltransferase | VRD$_{100}$PSSLAATLVALK | 582.337 | 582.328 | 3 | 64.51 |
| AAC72448.1 | Hydroperoxide reductase | CPAHN$_{123}$TLHGTSNFK | 865.905 | 865.894 | 2 | 41.41 |
| AAC71293.1 | Uncharacterized protein MG075 | NLFSVIGDILS$_{581}$ETNVNK | 675.685 | 675.684 | 3 | 35.59 |
| AAC71503.1 | Uncharacterized protein MG281 [Protein M] | QVPSFSGWSN$_{202}$TK | 514.570 | 514.576 | 3 | 26.14 |
| AAC71410.1 | Adhesin P1 [MgPa] | HPE$_{407}$WFDEGQAK | 516.567 | 516.560 | 3 | 33.72 |

*y*-ion directions. For GroEL, the predicted and observed MS[1] masses for the unmodified peptide were 1631.886 and 1631.860, respectively with a mass error of −15.932 ppm. The predicted and observed hexosylated peptide masses were both 1574.866. Fourteen distinct proteins were identified from membrane-enriched Mgen datasets with instances of hexosylation with -log10P>25 (Table 1). Each of these proteins are known surface proteins or are known to moonlight on the surface according to MoonProt, a database of known moonlighting proteins [29]. Numerous instances of hexosylation at other proteins were identified but did not meet the stringent criteria of −10logP > 25.

### *N*- and *O*-linked hexosylation of Syn3A and Mmc parent at canonical amino acids

The translation elongation factor Tu (EF-Tu) is among the most abundant proteins in the Syn3A proteome and is known to moonlight on the surface of many bacteria including mycoplasmas [30]. We have found hexosylation at Thr$_{159}$ and another instance of hexosylation at Tyr$_{161}$ of Syn3A EF-Tu peptide LLT$_{159}$EY$_{161}$DFDGEGAPVIR (Fig 3). The predicted and observed

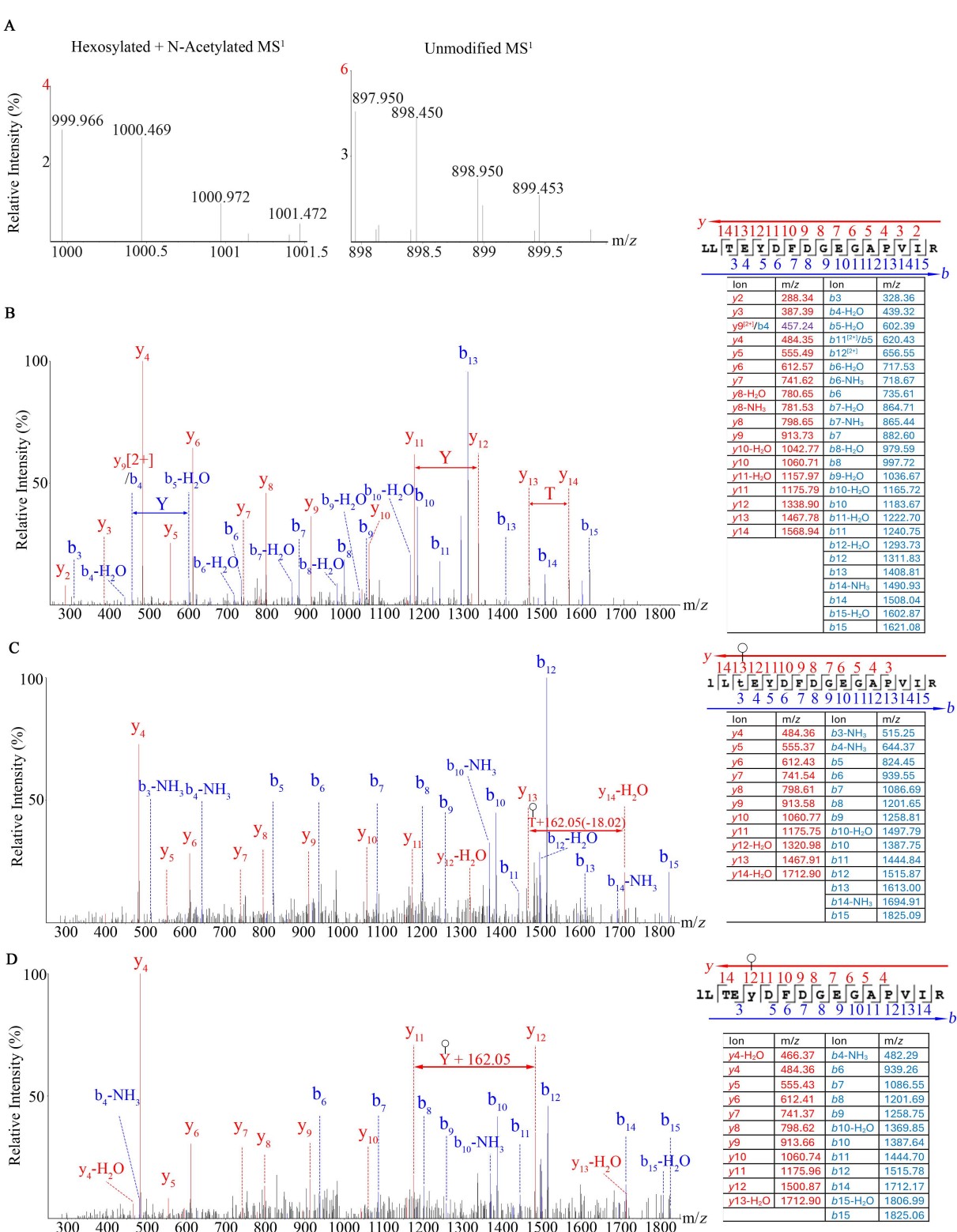

**Fig 3. Mass spectra of the (A) MS$^1$ precursors, (B) MS$^2$ of unmodified peptide, (C) MS$^2$ of the hexosylated Thr$_{159}$ peptide, and (D) MS$^2$ of hexosylated Tyr$_{161}$ peptide from the Syn3A EF-Tu (accession: AVX54645) protein.** Blue and red indicate *b* and *y*-ions, respectively. Circle designates hexose modification in accordance with the most recent Symbol Nomenclature for Glycans [31].

MS$^1$ masses for the unmodified peptides were 1759.898 and 1759.900 Da, respectively with a 1.136 ppm mass error while the predicted and observed hexosylated peptide MS$^1$ masses were 1999.962 and 1999.932 Da, respectively, with a mass error of −15.000 ppm. The MS$^1$ spectra of the hexosylated peptides also show acetylation on the N-terminal Lys$_{157}$ indicated by a + 102.016 Da mass shift when compared to the unmodified peptide MS$^1$. Both the modified and unmodified peptide precursor ions have a charge state of 2 (Fig 3A). The MS$^2$ spectra show 162.05 Da mass shift on Thr$_{159}$ (Fig 3C) and Tyr$_{161}$ (Fig 3D) when compared to the Thr$_{159}$ and Tyr$_{161}$ unmodified peptide (Fig 3B). Our searches identified 10 distinct proteins with hexosylation in Syn3A (Table 2) and 14 distinct proteins in Mmc (Table 3).

We have previously identified Gln as a valid substrate in the mycoplasma hexosylation reaction [4]. Unsurprisingly, we have identified *N*-linked hexosylation at Gln in Syn3A and Mmc. The peptide IPDQ$_{925}$VIDLNSATTDQK from the Syn3A uncharacterized lipoprotein (accession: AVX54801.1) showed hexosylation on the Gln$_{925}$ residue. The predicted and observed MS$^1$ mass for the unmodified peptide were both 879.450 Da. The MS$^1$ masses of the N-acetylated and hexosylated peptide were 981.481 and 981.493 Da, respectively. The MS$^2$ shows a Gln + 144.04 Da m/z shift, assigned as a hexosylated Gln residue with a water loss (S5 Fig).

## Non-canonical hexosylation of acidic amino acids

We unexpectedly identified the addition of hexoses to aspartic acid and glutamic acid. To our knowledge, this is the first instance of hexosylation of Glu and Asp reported in bacteria. We found hexosylation of PVE$_{216}$DVFTITGR in the Mmc EF-Tu protein (accession: A0A7Z7PLJ1). The predicted and observed MS$^1$ masses for the unmodified peptide were 617.327 and 617.326, respectively with a mass error of −1.620. The predicted and observed MS$^1$ masses for the modified peptide were 719.359 and 719.356, respectively with a mass error of −4.170 ppm. The MS$^2$ shows a 162.05 Da shift on the Glu$_{216}$ residue in both the *b*- and *y*-ion directions on the modified peptide that is not found on the unmodified peptide used for comparison (Fig 4).

Mgen's phosphate acetyltransferase (accession: AAC71521.1) was detected in our membrane-enriched material. We detected separate instances of hexosylation on the Asp$_{100}$ and Ser$_{102}$ residues in individual peptides from the same population of the VRD$_{100}$PS$_{102}$SLAATLVALK peptide sequence. The theoretical and observed MS$^1$ m/z for the detected hexosylated and acetylated peptide was 582.328 and 582.337, respectively, with a mass error of 15.455 ppm. The unmodified peptide used for comparison had an additional N-terminal Phe residue, the theoretical and observed MS$^1$ were both

**Table 2. Detected hexosylated Syn3A proteins.**

| Accession | Protein Name | Peptide | Observed m/z | Theoretical m/z | z | −10logP |
|---|---|---|---|---|---|---|
| AVX54645.1 | EF-Tu | LLT$_{159}$EYDFDGEGAPVIR | 999.966 | 999.981 | 2 | 73.49 |
| AVX54645.1 | EF-Tu | LLTEY$_{161}$DFDGEGAPVIR | 999.966 | 999.981 | 2 | 61.01 |
| AVX54671.1 | Dihydrolipoyl dehydrogenase* | ADIGEGLT$_{14}$EGTVAEVLVK | 669.021 | 669.012 | 3 | 60 |
| AVX54812.1 | L-lactate dehydrogenase | FY$_{234}$GIGACLTK | 667.326 | 667.321 | 2 | 69.2 |
| AVX54801.1 | Uncharacterized lipoprotein | IPDQ$_{925}$VIDLNSATTDQK | 981.493 | 981.481 | 2 | 56.97 |
| AVX54673.1 | Acetate kinase | HGIS$_{178}$YEYIVNK | 509.593 | 509.584 | 3 | 33 |
| AVX54670.1 | Branched-chain alpha keto acid dehydrogenase* | ADIGEGLT$_{14}$EGTVAEVLVK | 669.021 | 669.012 | 3 | 60 |
| AVX54645.1 | EF-Tu | PVE$_{216}$DVFTITGR | 719.366 | 719.359 | 2 | 32.21 |
| AVX54879.1 | Fatty acid binding protein | GILVD$_8$SAAVYDPAEFK | 949.981 | 949.968 | 2 | 65.49 |
| AVX54662.1 | Phosphopyruvate hydratase | WT$_{383}$AVVSHR | 559.274 | 559.286 | 2 | 55.57 |
| AVX54718.1 | Transketolase | DWDY$_{279}$DDFIIPDSVYK | 1047.969 | 1047.957 | 2 | 37.78 |
| AVX54783.1 | Fatty acid kinase subunit A | LEGIEY$_{196}$YVLNDQIVNK | 1057.543 | 1057.531 | 2 | 57.47 |

*Indicates a degenerate peptide sequence conserved between the AVX45670.1 and AVX54671.1 proteins.

**Table 3. Detected hexosylated Mmc proteins.**

| Accession | Protein Name | Peptide | Observed m/z | Theoretical m/z | z | −10logP |
|---|---|---|---|---|---|---|
| SRX69133.1 | Glycerol ABC transporter | PFDN$_{169}$ADTNALQINLR | 932.466 | 932.458 | 2 | 104.09 |
| SRX69148.1 | L-lactate dehydrogenase | VIASN$_{123}$PVDVITHVYQK | 649.018 | 649.014 | 2 | 43.34 |
| SRX69207.1 | DnaK | AELE$_{564}$QAMAQAAEFANK | 628.637 | 628.628 | 3 | 46.65 |
| SRX68848.1 | ABC transporter | VVADE$_{479}$PISALDVSIR | 894.480 | 894.467 | 2 | 89.22 |
| SRX68831.1 | EF-Tu | PVE$_{216}$DVFTITGR | 719.356 | 719.359 | 2 | 64.71 |
| SRX69115.1 | Lipoprotein | IPDQ$_{925}$VIDLNSATTDQK | 981.500 | 981.481 | 2 | 114.91 |
| SRX68746.1 | Alanine dehydrogenase | W*YS$_{298}$VPNIPGAVPR | 539.945 | 539.947 | 3 | 26.62 |
| SRX68910.1 | Dihydrolipoyl dehydrogenase | ADIGE$_{11}$GLTEGTVAEVLVK | 669.021 | 669.012 | 3 | 58.57 |
| SRX71236.1 | Uncharacterized protein | TVT$_{270}$DVDLSDPSLAINEGLLK | 576.795 | 576.798 | 4 | 42.19 |
| SRX69114.1 | Lipoprotein | DKPS$_{35}$ITDELSQK | 522.267 | 522.257 | 3 | 54.54 |
| SRX68912.1 | Acetate kinase | EDT$_{54}$LPDHEHAIQLILNK | 730.712 | 730.702 | 3 | 58.62 |
| SRX69241.1 | MOLPALP family lipoprotein | LNELGSSN$_{680}$INLTGLTDVAKV | 754.721 | 754.729 | 3 | 25.84 |
| SRX68847.1 | Peptide ABC transporter ATP-binding protein | RY$_{218}$FQIINLINK | 542.644 | 542.635 | 3 | 51.43 |
| SRX68730.1 | MurR/RpiR family transcriptional regulator | AIN$_{107}$ATDALIETNQVDK | 939.462 | 939.471 | 2 | 44.12 |

*Indicates a substitution from His to Trp

563.330. The similarities of the fragmentation patterns in the *b*- and *y*-ion directions increased our confidence of the two modified peptides having equivalent amino acid sequences, furthermore the $y_{10}^{2+}$ and $y_{11}^{2+}$ assignments found in the Ser$_{102}$-hexosylated peptide were not detected in the D$_{100}$-hexosylated peptide (Fig 5).

We also identified hexosylation of GILVD$_8$SAAVYDPAEFK in the Syn3A fatty-acid binding protein (accession: AVX54879.1). The theoretical and observed masses of the unmodified peptide were both 847.935 Da. The theoretical and observed masses of the modified peptide were 949.968 and 949.981 Da, respectively with a mass error of 13.685 ppm. The MS$^2$ shows a mass shift on the Asp$_8$ residue indicative of a hexose being attached (Fig 6).

## Discussion

The variable hexosylation system previously reported in *M. arthritidis* and *M. pulmonis* is conserved in the mycoplasmas with the smallest genomes, Mgen and the synthetic organism JCVI-Syn3A, as well as Syn3A's Mmc parent organism. In addition to the previously reported modification of serine, threonine, asparagine, and glutamine, we detected evidence of hexosylation on tyrosine and surprisingly, aspartic acid and glutamic acid residues. To our knowledge no glycosylation of the acidic amino acids has been reported in bacteria, but biochemically the acidic amino acids should be valid glycosyl acceptors (Fig 7) and *O*-GlcNAcylation of the acidic amino acids has been detected in human samples [32]. Many amino acids can be non-enzymatically glycated, however we have not detected mass shifts indicative of hexoses being attached to amino acids commonly reported to be glycated, such as lysine, arginine, and cysteine [33]. Our growth medium is supplemented with the disaccharide maltose. However, we did not detect disaccharides linked to amino acids as would be expected if glycation reactions were responsible for these observed modifications. Lastly, monosaccharides are substrates for glycation, whereas, as we showed previously [4,34], this hexosylation reaction utilizes oligosaccharides as the substrate. Therefore, we conclude that hexosylation in mycoplasmas occurs enzymatically.

The gene(s) responsible for mycoplasma hexosylation are unknown. We were unable to identify many candidates for hexosyltransferase(s) when searching mycoplasma proteomes using bioinformatic tools [2]. Given that Mgen and Syn3A both contain genomes under 600-kbp in length, it is unlikely that numerous glycosyltransferases conserved between the two organisms are catalyzing reactions for each amino acid being modified. Many biochemical reactions in mycoplasmas

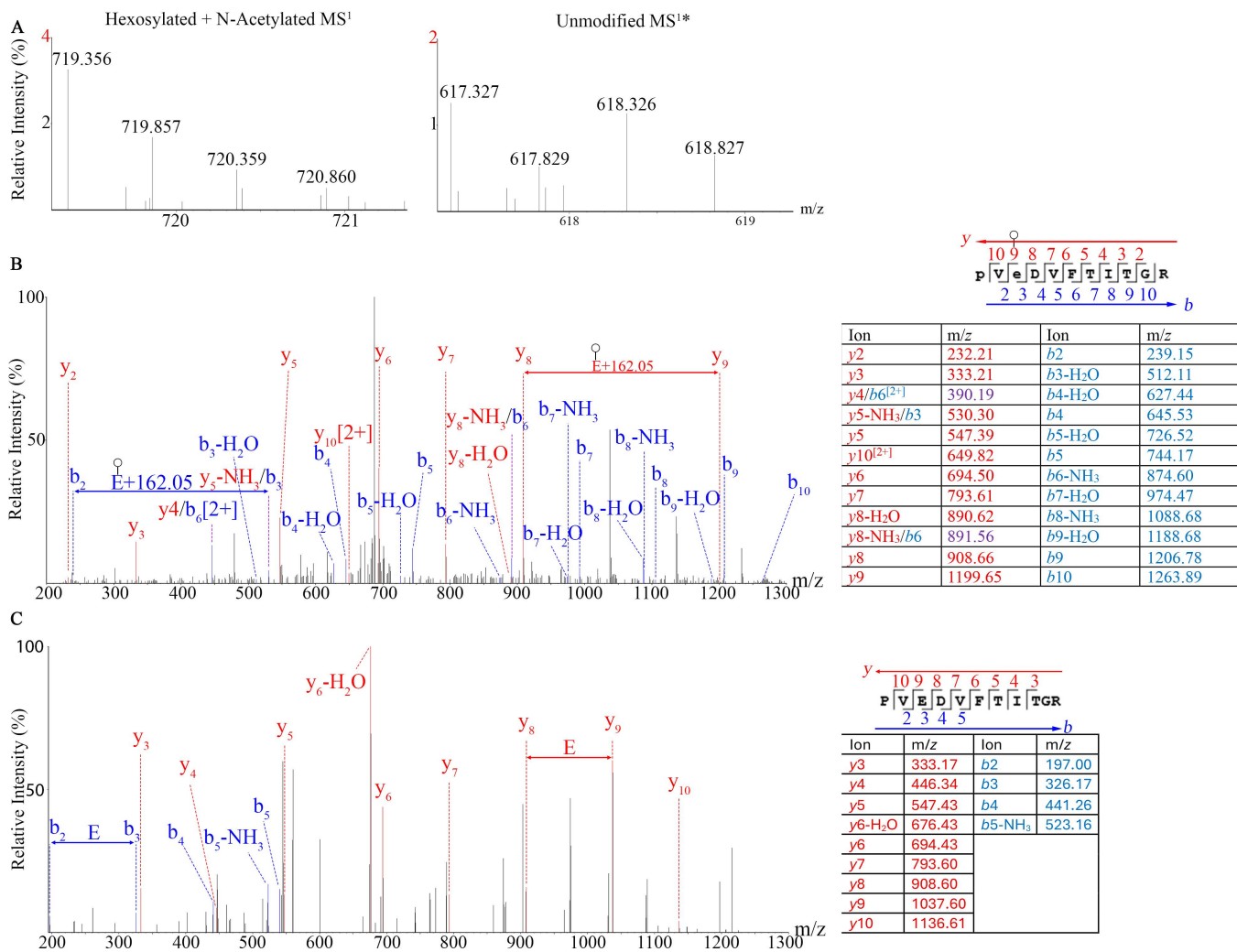

**Fig 4. Mass spectra of the (A) MS$^1$ precursors*, (B) MS$^2$ of a hexosylated Glu$_{216}$ residue and (C) MS$^2$ of nearly identical unmodified peptide from the GM12 EF-Tu protein (Uniprot accession: A0A7Z7PLJ1) protein.** *The unmodified peptide used for this comparison was detected with a shorter HPLC gradient compared to the modified peptide.

exhibit "leaky" characteristics. One profound example are mycoplasma aminoacyl-tRNA synthetases, which have a remarkably low fidelity when compared to other bacteria, resulting in up to 0.5% of incorrectly charged aminoacyl-tRNAs recruited to the translation complex [35,36]. Given the nature of mycoplasma enzymes, we suggest one or more currently undescribed glycosyltransferase(s) could be catalyzing these reactions. Given the requirement for oligosaccharides, the glycosyltransferase(s) responsible for these reactions may be viewed as a glycosidase involved in scavenging oligosac-charides from the host. This process is remarkably elegant but reasonable for an organism with extensive metabolic adap-tation and streamlining to a parasitic lifestyle, as host oligosaccharides would provide the energy required for the reaction and allow it to occur on the outside of the cell.

It has long been known that cytoplasmic enzymes are found on the surface with largely unknown physiological roles in many bacteria [37,38], including mycoplasmas [39]. These proteins are often referred to as moonlighting proteins. All hexosylated proteins identified in this study are known to exist on the surface of mycoplasmas, being annotated as

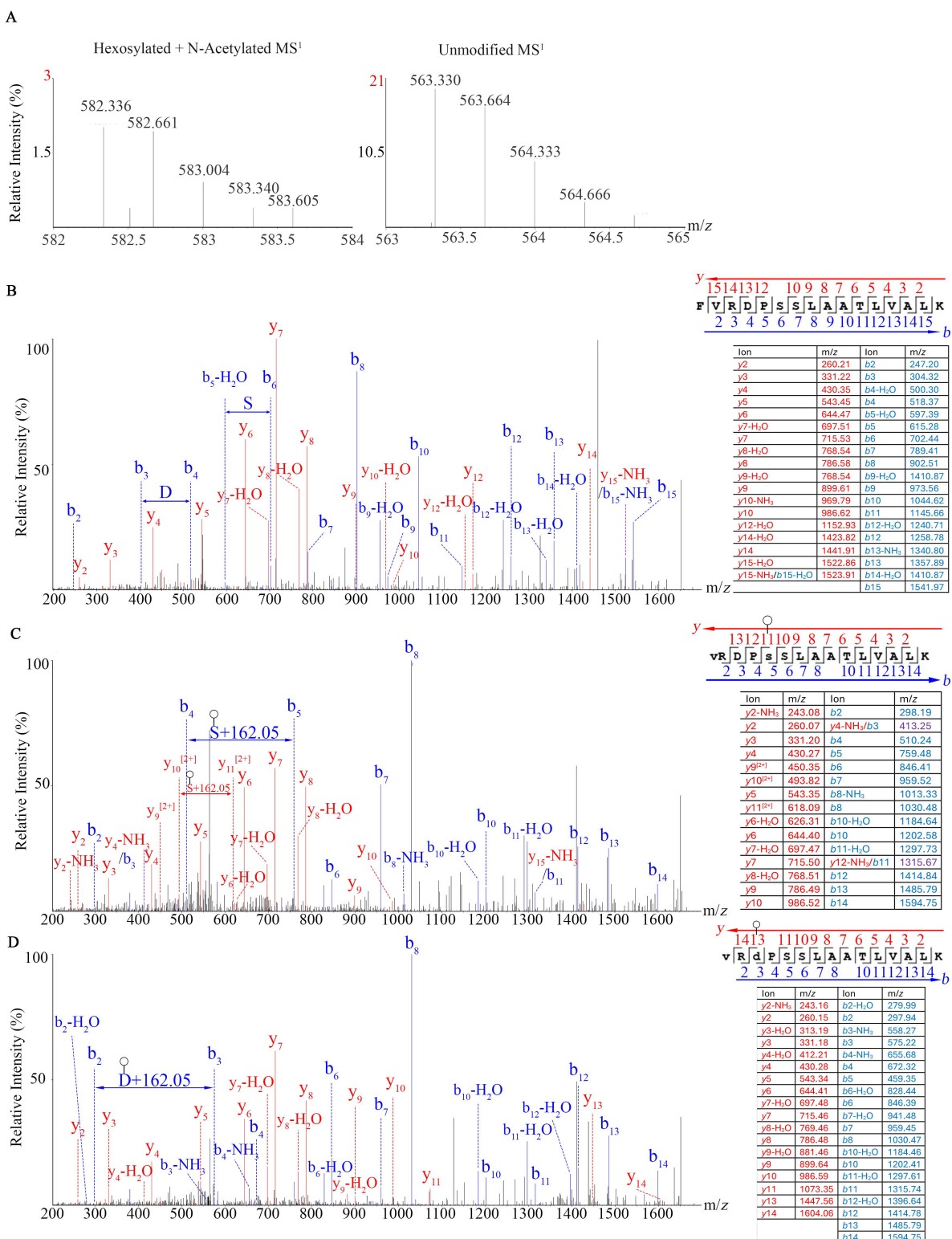

**Fig 5. Mass spectra of the (A) MS$^1$ precursors, (B) MS$^2$ of unmodified peptide, (C) MS$^2$ of the hexosylated D$_{100}$ peptide, and (D) MS$^2$ of the hexosylated Ser$_{102}$ peptide from the Mgen phosphate acetyltransferase (accession: AAC71521.1) protein.**

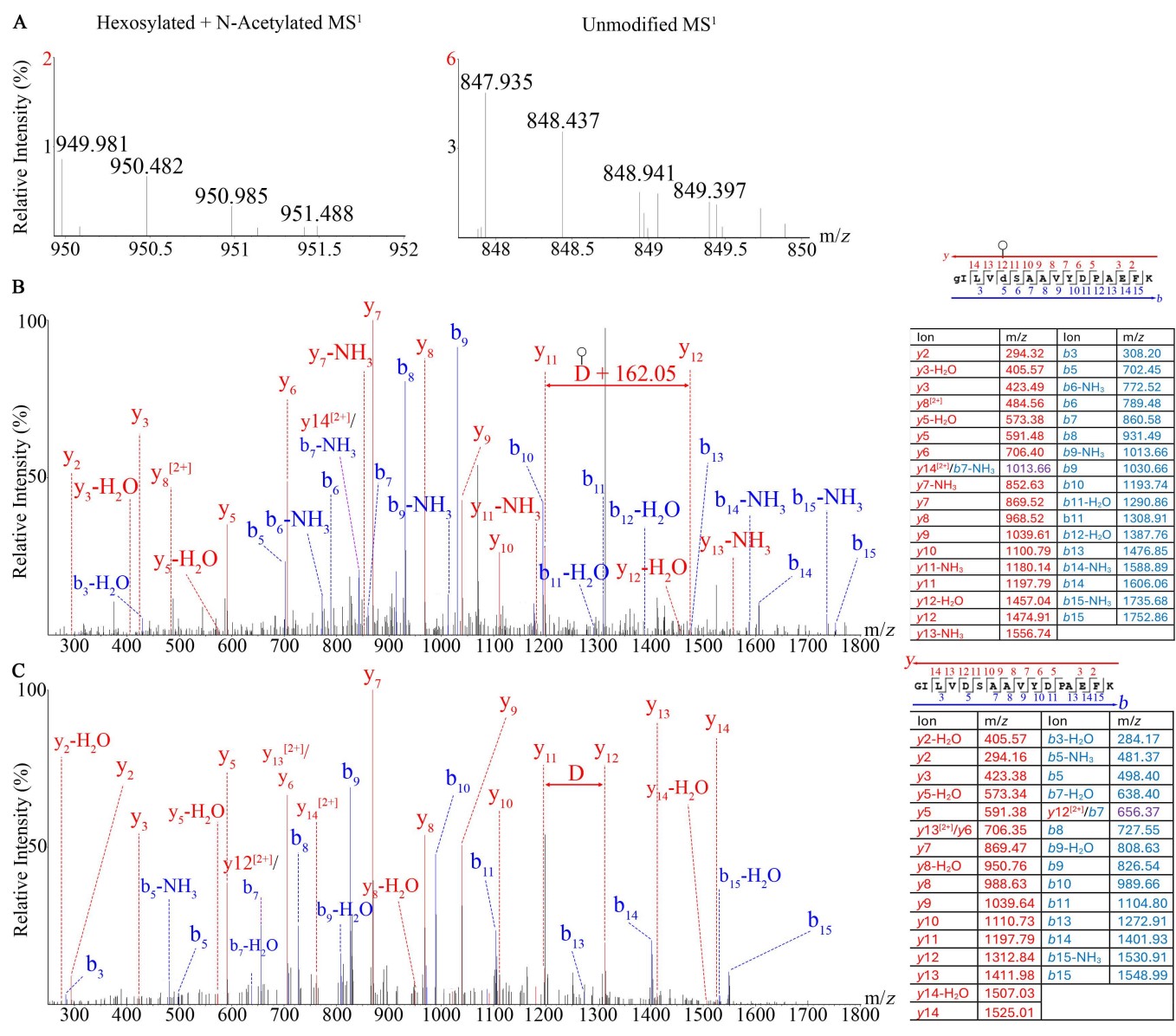

**Fig 6. Mass spectra of (A) MS¹ precursors, (B) hexosylated Glu₈ residue and (C) unmodified peptide from a Syn3A fatty acid-binding protein (accession: AVX54879.1) protein.**

lipoproteins or known moonlighting proteins. We hypothesize that structural accessibility determines which amino acids can be glycosylated. This hypothesis is supported by predicted structures of Mgen, Mmc and Syn3A through Alphafold 3. Our observed Syn3A EF-Tu glycosylation sites at Thr₁₅₉ and Tyr₁₆₁ residues are located on solvent-exposed loops (Fig 8). Likewise, the Mgen EF-Tu Ser₁₄₁ hexosylation site is readily accessible for modification according to its predicted structure (S6 Fig).

Monosaccharides are not utilized in mycoplasma hexosylation [2,34]. The growth medium used in this study contained 3 mg/mL of maltose, a disaccharide of glucose which would result in an overrepresentation of glucose attached to proteins identified in this study. Our GC-MS of gel-purified proteins show glucose as the predominant sugar in hexosylated

**Fig 7. The hypothesized mechanism for hexosylation in mycoplasmas.** In addition to the previously known *O*-linked (Ser, Thr) and *N*-linked (Asn, Gln) amino acid substrates, Tyr and the acidic amino acids (Asp, Glu) are available for glycosylation.

proteins. An appreciable concentration of mannose was also observed in the excised material (Fig 2). The presence of mannose can be explained by mycoplasmas scavenging the heavily mannosylated proteins from yeast extract found in the SP4 growth medium used in these experiments. In the context of colonization or infection *in vivo*, mycoplasmas would scavenge from animal oligosaccharides, suggesting that the variety of monosaccharides could be used for hexosylation as available from the host. The monosaccharide composition of PTMs on surface proteins in individual bacteria could be driven by the microenvironment in the host.

A newly synthesized protein should not be hexosylated, but over time hexosylation events would occur. Since the glycosyl donor is an exogenous oligosaccharide and the glycosyl acceptor does not have a consensus sequence, the variability of hexosylated proteins is remarkably high. This may explain the variability between the identified Mmc and Syn3A proteins that are hexosylated. We presume that with large enough proteomic datasets, the list of modified Mmc and Syn3A proteins would become much greater and highly similar. More quantitative analysis on such a large dataset could answer more questions about the variability of this phenomenon on protein identity and the specific residue of the modification site that is beyond the scope of this current study. Perhaps mycoplasma hexosylation is chaotic, with the glycosyl receptor's spatial availability and oligosaccharide glycosyl donor's identity (i.e., carbohydrate residues and bond linkages) acting as deterministic factors while perturbated by initial conditions at the mycoplasma cell surface.

A challenging component to our study is identifying the residue-specific glycosylation site due to the low abundance of hexosylated peptides in our samples. Our detection of hexosylation on the peptide level was confirmed using both the PEAKS and SEQUEST search algorithms with high confidence. Previous attempts to perform electron-transfer dissociation in these samples has not resulted in satisfactory fragmentation, perhaps due to the peptides in our samples being smaller in size and generally carrying lower charge states or the peptide sequences not containing many basic residues [40]. Furthermore, traditional glycosylation site identification methods such as treatment with PNGase F, Endo F, or Endo H will not work with these samples due to the requirement of an Asn residue to specifically cleave classical *N*-linked

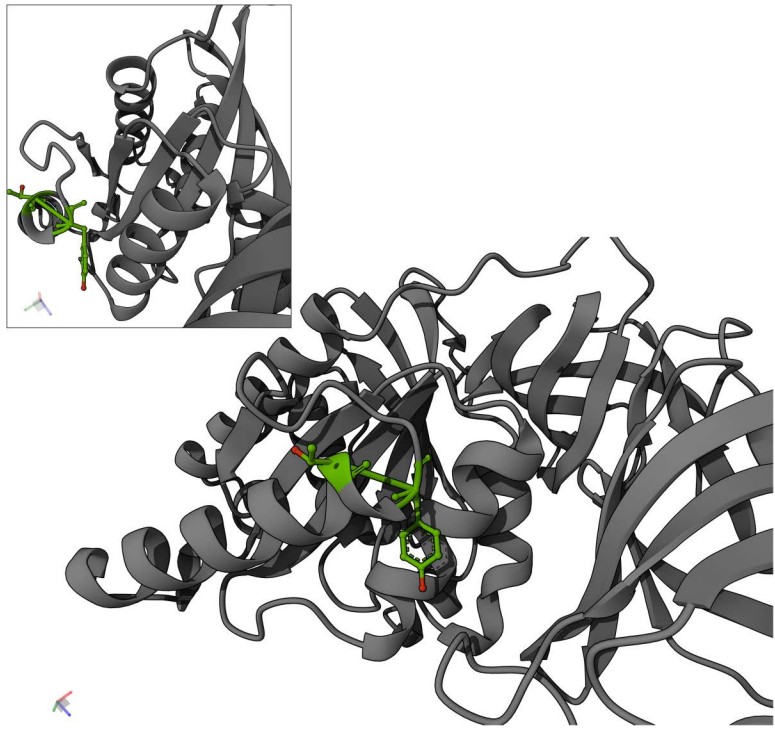

**Fig 8. Alphafold prediction of Syn3A Elongation Factor-Tu.** The location of Thr$_{159}$ and Tyr$_{161}$ show the hydroxyl groups that are *O*-glycosylated are accessible for modification. Inset shows a rotated view of the modification sites. Green shows the locations of Thr$_{159}$ and Tyr$_{161}$ (also depicted as ball-and-stick models), red shows the hydroxyl groups involved in *O*-bond formation.

glycans. Further developments in techniques to identify glycosylation sites could lead to increased confidence in the protein glycosylation sites identified here.

The presence of hexosylation in the minimal natural organism Mgen and the synthetic organism Syn3A implies essentiality even under axenic culturing conditions. Mycoplasmas are unable to synthesize many amino acids *de novo* because of genome streamlining to suit a parasitic lifestyle and instead scavenge host proteins for amino acids. Hence these organisms have an abundance of proteases that are presumed to be employed for scavenging amino acids and survival in the host [41,42]. PTMs can inhibit protease accessibility in the immediate area around the modification site [43]. Therefore, we previously proposed the protein hexosylation function is to protect the mycoplasma proteins from self-digestion from proteases [4]. However, in this study the frequency of glycosylation observed in Mgen and Syn3A appeared to be low, undermining the concept of a protective role. Another possible explanation for essentiality under axenic culture conditions is modulating of the cell surface's hydrophobicity, similar to the requirement for lipid glycosylation in these organisms [44]. This current study was performed under axenic conditions, but hexosylation would likely modulate a wide range of interactions with the animal host during infection such as host immune responses and adhesion. Further work is required to characterize the functional role of this conserved PTM, the gene(s) involved in this process, and potential applications of this phenomenon such as new drug targets against pathogenic mycoplasmas including Mgen and Mmc.

## Supporting information

**S1 Fig. Raw gel images.**
(PDF)

**S2 Fig. Gas chromatogram from all excised bands showing in Fig 1 showing the presence of glucose and mannose in each sample.**
(PDF)

**S3 Fig. MS$^1$ and MS$^2$ showing hexosylation on Asn$_{10}$ of Mgen acetate kinase (accession: AAC71582.1).**
(TIF)

**S4 Fig. MS$^1$ and MS$^2$ showing hexosylation on Asn$_{45}$ of Mgen chaperonin GroEL (accession: AAC71620.2).**
(TIF)

**S5 Fig. MS$^1$ and MS$^2$ showing hexosylation on a Syn3A uncharacterized lipoprotein (accession: AVX54801.1).**
(TIF)

**S6 Fig. Alphafold predicted structure of the Mgen EF-Tu Ser$_{141}$ hexosylation site.** Green shows the locations of Ser$_{141}$ (also depicted as ball-and-stick models), red shows the hydroxyl groups involved in *O*-bond formation.
(PDF)

## Acknowledgments

We would like to thank Dr. John Glass and Dr. Kim Wise (J. Craig Venter Institute, La Jolla, CA) for generously providing JCVI-Syn3A and fruitful correspondence. We would also like to thank Dr. Kyoko Kojima (UAB Mass Spectrometry/Proteomics Shared Resource) for performing the LC-MS/MS experiments in this manuscript.

## Author contributions

**Conceptualization:** John William Sanford, James Mobley, Kevin Dybvig, Thomas Prescott Atkinson, James Daubenspeck.

**Data curation:** James Mobley.

**Formal analysis:** John William Sanford, James Daubenspeck.

**Funding acquisition:** Thomas Prescott Atkinson.

**Investigation:** John William Sanford, James Mobley, James Daubenspeck.

**Methodology:** John William Sanford, James Mobley, Kevin Dybvig, James Daubenspeck.

**Resources:** James Mobley, Thomas Prescott Atkinson, James Daubenspeck.

**Supervision:** Kevin Dybvig, Thomas Prescott Atkinson.

**Validation:** John William Sanford, James Mobley, Kevin Dybvig, James Daubenspeck.

**Visualization:** John William Sanford, Kevin Dybvig, James Daubenspeck.

**Writing – original draft:** John William Sanford, James Mobley, Kevin Dybvig, Thomas Prescott Atkinson, James Daubenspeck.

**Writing – review & editing:** John William Sanford, James Mobley, Kevin Dybvig, Thomas Prescott Atkinson, James Daubenspeck.

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
