## [Decision Letter · Decision Letter 0]

4 Jun 2025

Dear Dr. Daubenspeck,

Thank you for submitting your manuscript to PLOS ONE. After careful consideration, we feel that it has merit but does not fully meet PLOS ONE’s publication criteria as it currently stands. Therefore, we invite you to submit a revised version of the manuscript that addresses the points raised during the review process.

We look forward to receiving your revised manuscript.

Kind regards,

Chih-Horng Kuo, Ph.D.

Academic Editor

PLOS ONE

Journal Requirements:

4. Please include captions for your Supporting Information files at the end of your manuscript, and update any in-text citations to match accordingly. Please see our Supporting Information guidelines for more information: http://journals.plos.org/plosone/s/supporting-information .

Additional Editor Comments:

Two experts in mycoplasmology found the study well-conceived, clearly presented, and of interest to the field. Please revise according to their comments and suggestions.

Reviewers' comments:

Reviewer's Responses to Questions

**Comments to the Author**

1. Is the manuscript technically sound, and do the data support the conclusions?

Reviewer #1: Yes

Reviewer #2: Yes

2. Has the statistical analysis been performed appropriately and rigorously?

Reviewer #1: N/A

Reviewer #2: Yes

3. Have the authors made all data underlying the findings in their manuscript fully available?

Reviewer #1: Yes

Reviewer #2: Yes

4. Is the manuscript presented in an intelligible fashion and written in standard English?

Reviewer #1: Yes

Reviewer #2: Yes

Reviewer #1: Sanford et al. have described hexosylation of membrane proteins in Mycoplasma genitalium, JCVI.Syn3A (a synthetically minimized organism), and Mycoplasma mycoides subsp. capri, the naturally occurring parent of the latter. This extends observations of similar modifications in two other species of mycoplasma. The modification target amino acids are non-canonical, including not only the usual amino acids but also aspartic acid and glutamic acid. Evidence is presented that the modification is enzymatic, using available hexoses perhaps in relation to their availability in the medium as components of oligosaccharides, as opposed to glycation. The mechanism, including the identity of the enzyme that catalyzes the transfer, is unidentified. The manuscript is very well-written and the experiments are appropriate to draw the conclusions that the authors have reached, although a small number of improvements can be made to enhance clarity.

Major issues:

1) Please describe why the specific bands in Figure 1 were selected for analysis.

2) Please indicate on all figures with spectra (including supplemental figures) the relationship between the peak and the identity of the modification it represents.

Minor issues:

3) Line 14: Is there a reason to say "may expand" instead of "expands"?

4) Line 31: "arthritidis"

5) Line 37: Although the phrase "smallest genome... capable of self-replication" is often thrown around, I honestly don't know what it means. All organisms need externally-derived material to replicate; they don't "know" whether that material is provided by a living cell or something else. The distinction is not "self-replication"; the distinction is that they can be grown without other living cells.

6) Line 39: Biofilms are structures composed of cells as well as material they secrete or capture. Biofilms cannot be described as being secreted.

7) Figure 7 panels A-C are commonly known structures. It is unclear to me what their inclusion adds.

8) Line 292: "but" instead of "however."

Reviewer #2: This manuscript reports on protein glycosylation in three Mycoplasma species with reduced genomes. The study has been conducted in a well-organized manner, and sufficient data have been presented to support the presence of glycosylation in the three Mycoplasmas.

The fact that glycosylation is observed even in Syn3A, which possesses a minimal genome, is highly interesting, and the role of glycosylation in maintaining Mycoplasma cells is of great interest. The discussion is also highly engaging and is considered to provide valuable insights for future research.

Minor comments:

Line 44

It is stated that the strain is “a highly virulent pathogen”; however, Syn3A was derived from Mycoplasma mycoides subsp. capri (Mmc), which is known to cause mild symptoms. Mmc differs from Mycoplasma mycoides subsp. mycoides small colony (Mmm), the causative agent of contagious bovine pleuropneumonia. Therefore, the term “highly” should be removed.

Line 65

It is recommended that the reasons for adding 3 mg/mL of maltose be briefly explained here. Although the reasons are clearly described in the Discussion section, it is not apparent when reading the Methods section alone.

Lines 70–71

It may be helpful to explain the reasons for this overnight step. This step is presumed to serve the purpose of inactivating protease activity; however, the need for such prolonged treatment may be due to irreversible inhibition or the requirement for the reagent to penetrate into the cells. Including such a description would be informative for readers and would aid in better understanding.

Line 162

The reasons for selecting bands A and B from among lots of protein bands should be provided.

Lines 165–166 and Figure 2

In Figure 2, is it possible to add labels on the peaks corresponding to glucose and mannose?

Tables 2 and 3

Regarding the comparison between Syn3A and its parental strain Mmc, many of the glycosylated proteins identified are different. As Syn3A is a genome-reduced strain, there are almost no additional genes compared to Mmc. Therefore, it would be expected that glycosylation detected in Syn3A should also be detectable in Mmc. The reasons for these differences should be discussed. Could this be due to the detection sensitivity or threshold settings in the MS analysis?

Figures 3–6

The resolution of the figures is low and the font size is too small, making it difficult to read the text in the figures. Please improve the figures by increasing both the font size and resolution. As figure readability also depends on the final published size, the font size and resolution should be adjusted accordingly.

**Do you want your identity to be public for this peer review?** For information about this choice, including consent withdrawal, please see our Privacy Policy

Reviewer #1: No

Reviewer #2: No

---

## [Author Response · Author response to Decision Letter 1]

9 Jul 2025

July 8, 2025

Editor, PLoS ONE

1160 Battery Street

Koshland Building East, Suite 100

San Francisco, CA 94111

Dear Sir/Ms:

We thank the reviewers for their careful review of our manuscript. Please find below the detailed responses to their critiques:

Reviewer #1:

Major issues:

1) Please describe why the specific bands in Figure 1 were selected for analysis.

We have expanded the sentence starting at line 184 to say: “Two sets of bands from gel regions that have been previously shown to contain modified proteins in M. arthritidis and M. pulmonis were excised from the Coomassie-stained gel from each organism.”

2) Please indicate on all figures with spectra (including supplemental figures) the relationship between the peak and the identity of the modification it represents.

We have adjusted our figure’s visual design to more clearly show which peaks represent the modified amino acid.

Minor issues:

3) Line 14: Is there a reason to say "may expand" instead of "expands"?

We have removed ‘may’ from the sentence.

4) Line 31: "arthritidis"

We have corrected that spelling mistake and appreciate the reviewer noting it.

5) Line 37: Although the phrase "smallest genome... capable of self-replication" is often thrown around, I honestly don't know what it means. All organisms need externally-derived material to replicate; they don't "know" whether that material is provided by a living cell or something else. The distinction is not "self-replication"; the distinction is that they can be grown without other living cells.

We concur that this is a difficult phrase to word appropriately. We have changed it to “Mycoplasma genitalium (Mgen) is a human urogenital pathogen that has streamlined its 580-kbp genome under natural selective pressures resulting in the smallest genome of any identified organism capable of axenic culture.”

6) Line 39: Biofilms are structures composed of cells as well as material they secrete or capture. Biofilms cannot be described as being secreted.

To improve the accuracy of our wording, we have changed it to “produces extracellular polymeric substances to form biofilms.”

7) Figure 7 panels A-C are commonly known structures. It is unclear to me what their inclusion adds.

We concur that panels A-C were superfluous and have streamlined figure 7 to just show the hypothesized mechanism.

8) Line 292: "but" instead of "however."

We have incorporated this change.

Reviewer #2:

Minor comments:

1) Line 44: It is stated that the strain is “a highly virulent pathogen”; however, Syn3A was derived from Mycoplasma mycoides subsp. capri (Mmc), which is known to cause mild symptoms. Mmc differs from Mycoplasma mycoides subsp. mycoides small colony (Mmm), the causative agent of contagious bovine pleuropneumonia. Therefore, the term “highly” should be removed.

We appreciate the reviewer suggesting this phrasing implies Syn3A descends from Mmm, we have therefore removed ‘highly’ as suggested.

Line 65: It is recommended that the reasons for adding 3 mg/mL of maltose be briefly explained here. Although the reasons are clearly described in the Discussion section, it is not apparent when reading the Methods section alone.

We have expanded the sentence to include “3 mg/mL of maltose added to the medium to act as the required oligosaccharide substrate for mycoplasma hexosylation” to improve readability.

Lines 70–71: It may be helpful to explain the reasons for this overnight step. This step is presumed to serve the purpose of inactivating protease activity; however, the need for such prolonged treatment may be due to irreversible inhibition or the requirement for the reagent to penetrate into the cells. Including such a description would be informative for readers and would aid in better understanding.

To clarify the purpose of this step, we have changed the wording of the paragraph starting at line 71. We thank the reviewer for noting the ambiguity of this section of the methods section and hope this has clarified the purpose of the overnight equilibration step.

Line 162: The reasons for selecting bands A and B from among lots of protein bands should be provided.

We have expanded the sentence starting at line 173 to say: “Two sets of bands from gel regions that have been previously shown to contain modified proteins in M. arthritidis and M. pulmonis were excised from the Coomassie-stained gel from each organism.”

Lines 165–166 and Figure 2: In Figure 2, is it possible to add labels on the peaks corresponding to glucose and mannose?

We have added labels to show which peaks are which glycans on Figure 2, we thank the reviewer for improving the clarity of our figure.

Tables 2 and 3: Regarding the comparison between Syn3A and its parental strain Mmc, many of the glycosylated proteins identified are different. As Syn3A is a genome-reduced strain, there are almost no additional genes compared to Mmc. Therefore, it would be expected that glycosylation detected in Syn3A should also be detectable in Mmc. The reasons for these differences should be discussed. Could this be due to the detection sensitivity or threshold settings in the MS analysis?

We appreciate the reviewer asking this astute question. We believe the variability between the modified proteins in both organisms is due to our current sample size detecting few hexosylation events. We propose that with a large enough sample, we would see more overlap between proteins conserved between the two organisms with variability in glycosylation sites, assuming our model is correct. We have added a paragraph starting at line 403 conjecturing why this is the case.

Figures 3–6

The resolution of the figures is low and the font size is too small, making it difficult to read the text in the figures. Please improve the figures by increasing both the font size and resolution. As figure readability also depends on the final published size, the font size and resolution should be adjusted accordingly.

We have increased the resolution and adjusted the fonts and visual design of our slides to be significantly more readable. We hope this is much easier to read and follow.

---

## [Decision Letter · Decision Letter 1]

16 Jul 2025

Dear Dr. Daubenspeck,

Thank you for submitting your manuscript to PLOS ONE. After careful consideration, we feel that it has merit but does not fully meet PLOS ONE’s publication criteria as it currently stands. Therefore, we invite you to submit a revised version of the manuscript that addresses the points raised during the review process.

We look forward to receiving your revised manuscript.

Kind regards,

Chih-Horng Kuo, Ph.D.

Academic Editor

PLOS ONE

Journal Requirements:

Additional Editor Comments:

In the abstract, the sentence referring to the “smallest genome… capable of self-replication” is inconsistent between the Response and the Main Text File.

There is no need to emphasize the phrase “smallest genome”, as this point is not central to the current study and may be revised as new organisms with smaller genomes are discovered in the future.

In addition, the version in the Response describes Mycoplasma genitalium as “a human urogenital pathogen that has streamlined its 580-kbp genome under natural selective pressures”. I do not agree with this use of the active voice, which incorrectly implies agency in an evolutionary process.

I suggest revising the sentence as follows:

“We report here that this surface protein hexosylation system is conserved in Mycoplasma genitalium, a human urogenital pathogen with a 580-kbp genome that can be cultured axenically.”

This manuscript can be accepted pending this final revision.

Reviewers' comments:

Reviewer's Responses to Questions

**Comments to the Author**

Reviewer #1: (No Response)

Reviewer #2: All comments have been addressed

2. Is the manuscript technically sound, and do the data support the conclusions?

Reviewer #1: Yes

Reviewer #2: Yes

3. Has the statistical analysis been performed appropriately and rigorously?

Reviewer #1: Yes

Reviewer #2: N/A

4. Have the authors made all data underlying the findings in their manuscript fully available?

Reviewer #1: Yes

Reviewer #2: Yes

5. Is the manuscript presented in an intelligible fashion and written in standard English?

Reviewer #1: Yes

Reviewer #2: Yes

Reviewer #1: My concerns have all been addressed, except that the "self-replicating" statement still appears in the abstract.

Reviewer #2: (No Response)

**Do you want your identity to be public for this peer review?** For information about this choice, including consent withdrawal, please see our Privacy Policy

Reviewer #1: No

Reviewer #2: No

---

## [Author Response · Author response to Decision Letter 2]

16 Jul 2025

Editor, PLoS ONE

1160 Battery Street

Koshland Building East, Suite 100

San Francisco, CA 94111

Dear Sir/Ms:

We thank the reviewers for their careful review of our manuscript’s revisions. Please find below the detailed responses to their critiques:

Reviewer #1:

Major issues:

In the abstract, the sentence referring to the “smallest genome… capable of self-replication” is inconsistent between the Response and the Main Text File.

There is no need to emphasize the phrase “smallest genome”, as this point is not central to the current study and may be revised as new organisms with smaller genomes are discovered in the future.

In addition, the version in the Response describes Mycoplasma genitalium as “a human urogenital pathogen that has streamlined its 580-kbp genome under natural selective pressures”. I do not agree with this use of the active voice, which incorrectly implies agency in an evolutionary process.

I suggest revising the sentence as follows:

“We report here that this surface protein hexosylation system is conserved in Mycoplasma genitalium, a human urogenital pathogen with a 580-kbp genome that can be cultured axenically.”

This manuscript can be accepted pending this final revision.

We sincerely appreciate Reviewer 1’s attention to detail and mindfulness of clear language through both rounds of peer review and have therefore incorporated their suggested sentence into the abstract.

---

## [Editor Report · Decision Letter 2]

18 Jul 2025

Surface protein glycosylation conserved in the human pathogen Mycoplasma genitalium and retained in the synthetic organism JCVI-Syn3A

PONE-D-25-10974R2

Dear Dr. Daubenspeck,

We’re pleased to inform you that your manuscript has been judged scientifically suitable for publication and will be formally accepted for publication once it meets all outstanding technical requirements.

Kind regards,

Chih-Horng Kuo, Ph.D.

Academic Editor

PLOS ONE
---

## [Editor Report · Acceptance letter]

PONE-D-25-10974R2

PLOS ONE

Dear Dr. Daubenspeck,

I'm pleased to inform you that your manuscript has been deemed suitable for publication in PLOS ONE. Congratulations! Your manuscript is now being handed over to our production team.

Kind regards,

on behalf of

Dr. Chih-Horng Kuo

Academic Editor

PLOS ONE